# Bioinspired extracellular vesicles embedded with black phosphorus for molecular recognition-guided biomineralization

Yingqian Wang[1,5], Xiaoxia Hu[1,5], Lingling Zhang[2,3,5], Chunli Zhu[1,4], Jie Wang [1], Yingxue Li[1], Yulan Wang[2], Can Wang[2], Yufeng Zhang [2,3] & Quan Yuan [1,4]

Extracellular vesicles (EVs) are involved in the regulation of cell physiological activity and the reconstruction of extracellular environment. Matrix vesicles (MVs) are a type of EVs released by bone-related functional cells, and they participate in the regulation of cell mineralization. Here, we report bioinspired MVs embedded with black phosphorus (BP) and functionalized with cell-specific aptamer (denoted as Apt-bioinspired MVs) for stimulating biomineralization. The aptamer can direct bioinspired MVs to targeted cells, and the increasing concentration of inorganic phosphate originating from BP can facilitate cell biomineralization. The photothermal effect of the Apt-bioinspired MVs can also promote the biomineralization process by stimulating the upregulated expression of heat shock proteins and alkaline phosphatase. In addition, the Apt-bioinspired MVs display outstanding bone regeneration performance. Our strategy provides a method for designing bionic tools to study the mechanisms of biological processes and advance the development of medical engineering.

[1] Key Laboratory of Analytical Chemistry for Biology and Medicine (Ministry of Education), College of Chemistry and Molecular Sciences, Wuhan University, Wuhan 430072, China. [2] State Key Laboratory Breeding Base of Basic Science of Stomatology (Hubei-MOST) and Key Laboratory of Oral Biomedicine, Ministry of Education, School and Hospital of Stomatology, Wuhan University, Wuhan 430079, China. [3] Medical Research Institute, School of Medicine, Wuhan University, Wuhan 430071, China. [4] Molecular Science and Biomedicine Laboratory, State Key Laboratory of Chemo/Biosensing and Chemometrics, College of Chemistry and Chemical Engineering, Hunan University, Changsha 410082, China. [5] These authors contributed equally: Yingqian Wang, Xiaoxia Hu, Lingling Zhang. Correspondence and requests for materials should be addressed to J.W. (email: wangjie077@whu.edu.cn) or to Y.Z. (email: zyf@whu.edu.cn) or to Q.Y. (email: yuanquan@whu.edu.cn)

Extracellular vesicles (EVs) have been recognized as critical mediators of intercellular communication[1,2]. They are involved in the regulation of cell physiological activity and reconstruction of the extracellular environment, thereby influencing various physiological and pathological processes[3]. Matrix vesicles (MVs) are a type of EVs generated from bone-related functional cells[4]. Considerable evidence indicates that MVs participate in the initial step of biomineralization[5], and they are particularly critical for regulating cell mineralization. Bioinspired MVs can be engineered to serve as effective tools for the regulation of mineralization-related biological behavior and biomineralization-guided tissue engineering. The abundant inorganic phosphates within MVs can bind with calcium ion, thus leading to the deposition of mineral crystals that can support continuous crystal proliferation and further promote in vivo biomineralization[4,6–9]. Previous studies have also demonstrated that increasing concentration of inorganic phosphate within and around MVs facilitates cell biomineralization[10–13]. In view of this, it is pivotal to give these bioinspired MVs the capability of supplying sufficient inorganic phosphate for the biomineralization process. Additionally, endowing the bioinspired MVs with targeted recognition functionality can be a powerful strategy for improving the efficiency of biomineralization. Consequently, designing bioinspired MVs that can self-produce inorganic phosphates and specifically act on targeted cells or tissues can open a promising path for studying mineralization-related biological processes and advancing bioinspired mineralization-guided medical engineering.

Black phosphorus (BP), an emerging two-dimensional metal-free layered material, has recently attracted much attention in the electronic and photoelectric field owing to its unique layered structure and thickness-dependent bandgap[14–16]. Notably, due to its good biocompatibility, BP has been explored for potential biomedical applications[17,18]. BP can degrade in the physiological environment and eventually yield nontoxic phosphate anion, which can be the resource for mineralization[19,20]. Previous studies demonstrated that phosphorus-rich materials can stimulate mineralization and bone regeneration by increasing the local concentration of phosphate ions[21–23]. Furthermore, ultrasmall BP nanosheets with sizes of several nanometers (usually named as BP quantum dots, BPQDs) have high photothermal conversion efficiency and thus can be used as a promising photothermal agent[24]. Previous researches have indicated that hyperthermia promotes biomineralization by stimulating the up-regulated expression of proteins, including alkaline phosphatase (ALP) and heat shock proteins (HSP)[25–29], thus increasing the generation of mineral crystals and realizing bone reconstruction[29–31]. These special properties render BPQDs promising biomaterials for stimulating biomineralization and further promoting mineralization-guided biological behavior regulation and medical engineering.

In this paper, cell-targeting aptamer-modified bioinspired MVs (denoted as Apt-bioinspired MVs) are designed for promoting cell biomineralization (Fig. 1). The aptamer can guide the bioinspired MVs to targeted cells, where inorganic phosphate originating from BPQDs can coordinate with calcium ion to generate calcium phosphate for mineralization. In addition, the photothermal effect-induced hyperthermia can also promote the production of mineral crystals. Thus, Apt-bioinspired MVs-induced mineralization is expected to be achieved. The mineralization of osteoblasts is employed as a model for investigating the ability of our designed Apt-bioinspired MVs to promote mineralization. The Apt-bioinspired MVs can target the osteoblast-rich area, and inorganic phosphate originating from BPQDs coupled with hyperthermia can facilitate the biomineralization of osteoblasts. These Apt-bioinspired MVs display advantages of targeted action on osteoblasts and self-production of inorganic phosphates. Additionally, the Apt-bioinspired MVs have good performance in promoting mineralization-induced bone regeneration. Taking advantage of this bioinspired design, our strategy puts forward a potential pathway to design functional bioinspired materials for regulating biological function and advancing the development of biological medical engineering.

## Results

**Characterization of the Apt-bioinspired MVs.** The preparation of the Apt-bioinspired MVs is illustrated in Fig. 2a. BPQDs are first encapsulated in poly (lactic-co-glycolic acid) nanoparticles (denoted as PLGA NPs) to obtain the bioinspired MVs[19]. Then, surface modification with osteoblast-targeting aptamers[32] is performed to produce the Apt-bioinspired MVs. The scanning electron microscopy (SEM) image shows the high-yield synthesis of Apt-bioinspired MVs (Fig. 2b). Transmission electron microscopy (TEM) image shows that the Apt-bioinspired MVs exhibit uniform spherical shape with an average diameter of ~100 nm (Fig. 2c). Energy-dispersive X-ray (EDX) analysis further shows the presence of P in the developed Apt-bioinspired MVs, indicating that the BPQDs were successfully loaded into the MVs (Fig. 2d). The loading efficiency of BPQDs in the MVs is determined to be about 23.4% according to the results of EDX analysis. The encapsulation efficiencies of BPQDs and aptamers by the MVs are 95.7% and 16.0%, respectively. Atomic force microscopy (AFM) study also shows a spherical morphology with average diameter of ~180 nm, and the measured height is about 85 nm (Fig. 2e, f). The diameter is larger than that determined by TEM and SEM, which may be attributed to the deformation of the Apt-bioinspired MVs caused by adsorption of mica substrate. Owing to the excellent photothermal conversion efficiency, BPQDs are considered to be promising photothermal agents[24]. The temperature of the Apt-bioinspired MVs solution rapidly increases upon NIR irradiation, and the photothermal performance is consistent during temperature elevation and photothermal cycle, demonstrating that the Apt-bioinspired MVs can serve as potential photothermal agents (Supplementary Fig. 12). Since long-term exposure to physiological medium is necessary in clinical applications, the photothermal stability of Apt-bioinspired MVs dispersed in PBS was investigated by comparison with bare BPQDs (Fig. 2g). After a 7-day dispersion period, the photothermal performance of bare BPQDs deteriorated over time, indicating that the bare BPQDs degrade rapidly in the humid environment. However, the photothermal performance of the Apt-bioinspired MVs decreased much more slowly, confirming that BPQDs in the Apt-bioinsired MVs have better stability than the bare BPQDs.

The cytotoxicity of Apt-bioinspired MVs in vitro was evaluated using a fluorescent live/dead assay. Live cells were stained with calcein AM and showed green fluorescence, while dead cells were stained with propidium iodide (PI) and exhibited red fluorescence. As shown in Fig. 2h, few red osteoblasts are observed even at a high concentration of Apt-bioinspired MVs at 100 ppm. Furthermore, after irradiation with a NIR laser for 30 s, there are few dead osteoblasts in all groups with different concentrations of Apt-bioinspired MVs. These results demonstrate that the Apt-bioinspired MVs exhibit high biocompatibility. The in vivo toxicology of the Apt-bioinspired MVs was investigated systematically. Healthy female mice were randomly divided into six groups and subjected to different conditions, including: (1) intravenous injection with Apt-bioinspired MVs, (2) intravenous injection with Apt-bioinspired MVs and irradiation with an 808 nm laser for 10 min, (3) intravenous injection with bioinspired MVs, (4) intravenous injection with bioinspired MVs and

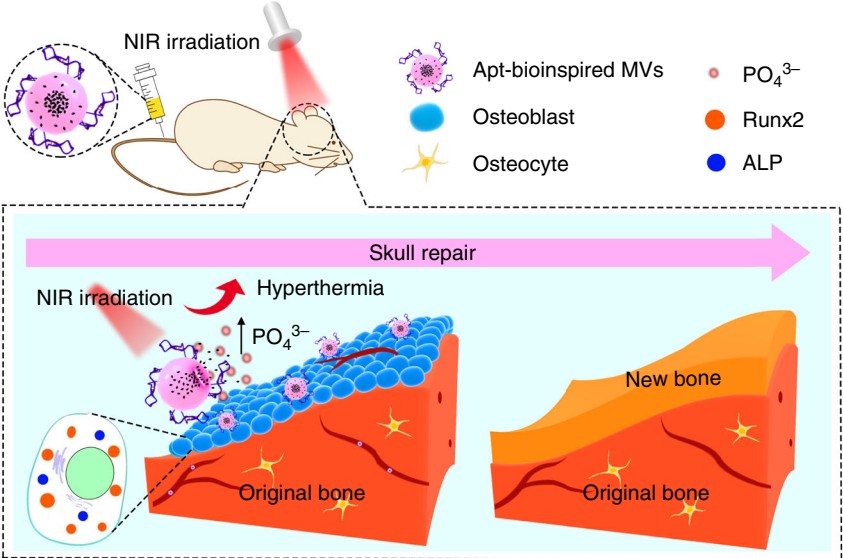

**Fig. 1** Overview of Apt-bioinspired MVs in cell mineralization. ALP, alkaline phosphatase; Runx2, runt-related transcription factor 2

irradiation with an 808 nm laser for 10 min, (5) intravenous injection with PBS, and (6) intravenous injection with PBS and irradiation with an 808 nm laser for 10 min. The hematological and blood biochemical analysis were carried out 28 days post injection. The standard hematology markers, including white blood cell count (WBC) and red blood cell count (RBC) were measured. As shown in Fig. 2i, the levels of WBC in these six groups are within the normal range. Also, the parameters of RBC in these six groups all appear to be normal (Fig. 2j). These results confirm that the Apt-bioinspired MVs do not cause infection nor inflammation in the treated mice, regardless of NIR irradiation. Furthermore, blood biochemical analyses were performed and the total protein (TP) and blood urea nitrogen (BUN) were examined. As shown in Fig. 2k, l, the levels of TP and BUN in all treated groups appear to be normal. Hence, the Apt-bioinspired MVs and their photothermal behavior have no side effects on the blood chemistry of mice. Taken together, no obvious toxicity is observed from the Apt-bioinspired MVs, demonstrating that they have good biocompatibility.

**Targeting ability of the Apt-bioinspired MVs.** DNA aptamer was conjugated to bioinspired MVs for recognition of the osteoblasts in bone region. Flow cytometry was used to assay the binding of osteoblast-targeting aptamer with osteoblasts. Osteoblasts incubated with Cy5-labeled aptamer display an obvious fluorescence shift (Fig. 3a), whereas no obvious fluorescence change is observed for rat dendritic cells incubated with Cy5-labeled aptamer (Fig. 3b), confirming that the aptamer exhibits much stronger binding affinity to rat osteoblasts compared to random cells. Furthermore, when the aptamer is incubated with whole bone marrow cells, an obvious fluorescence increase is produced (Fig. 3c), which can be ascribed to the osteoblasts in whole bone marrow. Considering that the osteoblasts are differentiated from mesenchymal cells and osteoblast precursors, flow cytometry was also used to assay the binding of the aptamer with mesenchymal cells and osteoblast precursor 3T3 cells. The mesenchymal cells and 3T3 cells incubated with Cy5-labeled aptamer display an obvious fluorescence shift (Fig. 3d, e), suggesting the co-expression of target protein on the surface of osteoblasts, osteoblast precursors, and mesenchymal cells. These results indicate that the Apt-bioinspired MVs can bind to many bone-regeneration-related cells, which can promote the

biomineralization process. The targeting function of the Apt-bioinspired MVs was further examined. Since the BPQDs in Apt-bioinspired MVs can induce hyperthermia under NIR irradiation, the binding of Apt-bioinspired MVs with osteoblasts was investigated by the infrared thermal wave imaging based on the photothermal effect (Fig. 3f). As shown in Fig. 3g, when the rat osteoblasts are incubated with Apt-bioinspired MVs, the average temperature of the plate increases to 39.9 °C after NIR irradiation for 30 s. After washing three times, the average temperature increases to 39.0 °C after irradiation for 30 s, indicating that the Apt-bioinspired MVs integrate tightly with osteoblasts. However, when the rat osteoblasts are incubated with bioinspired MVs without aptamer modification, the average temperature of the plate (36.3 °C) is close to the original temperature (35.3 °C) after washing, suggesting that bioinspired MVs cannot target osteoblasts. Similarly, when the dendritic cells are incubated with Apt-bioinspired MVs or bioinspired MVs, only slight enhancement of temperature is observed after washing, demonstrating that the Apt-bioinspired MVs have weak binding affinity to dendritic cells. The above results indicate that the Apt-bioinspired MVs can bind tightly with osteoblasts. As a result, the Apt-bioinspired MVs exhibit good selectivity and affinity to the rat osteoblasts, demonstrating that the aptamer provides a powerful tool for targeted delivery.

**In vitro biomineralization guided by Apt-bioinspired MVs.** Phosphorus is a key component of mineral crystals, and hyperthermia can promote the biomineralization process[25–31,33]. With the special capability to provide phosphate and hyperthermia, the Apt-bioinspired MVs were further studied for biomineralization purposes. The mechanism of osteoblast biomineralization mediated by the Apt-bioinspired MVs is illustrated in Fig. 4a. The generation of phosphates by the degradation of BPQDs within the Apt-bioinspired MVs was investigated (Supplementary Fig. 16). Gradually increased concentrations of phosphates in water is observed, and the concentration of phosphates reaches over 1 mg $L^{-1}$ after 20 days, showing the good ability of Apt-bioinspired MVs in continuously providing phosphates for biomineralization. The rat osteoblasts were incubated with Apt-bioinspired MVs (denoted as Apt-bioinspired MVs group), bioinspired MVs (denoted as bioinspired MVs group) and PLGA NPs (denoted as PLGA group) to investigate the biomineralization of

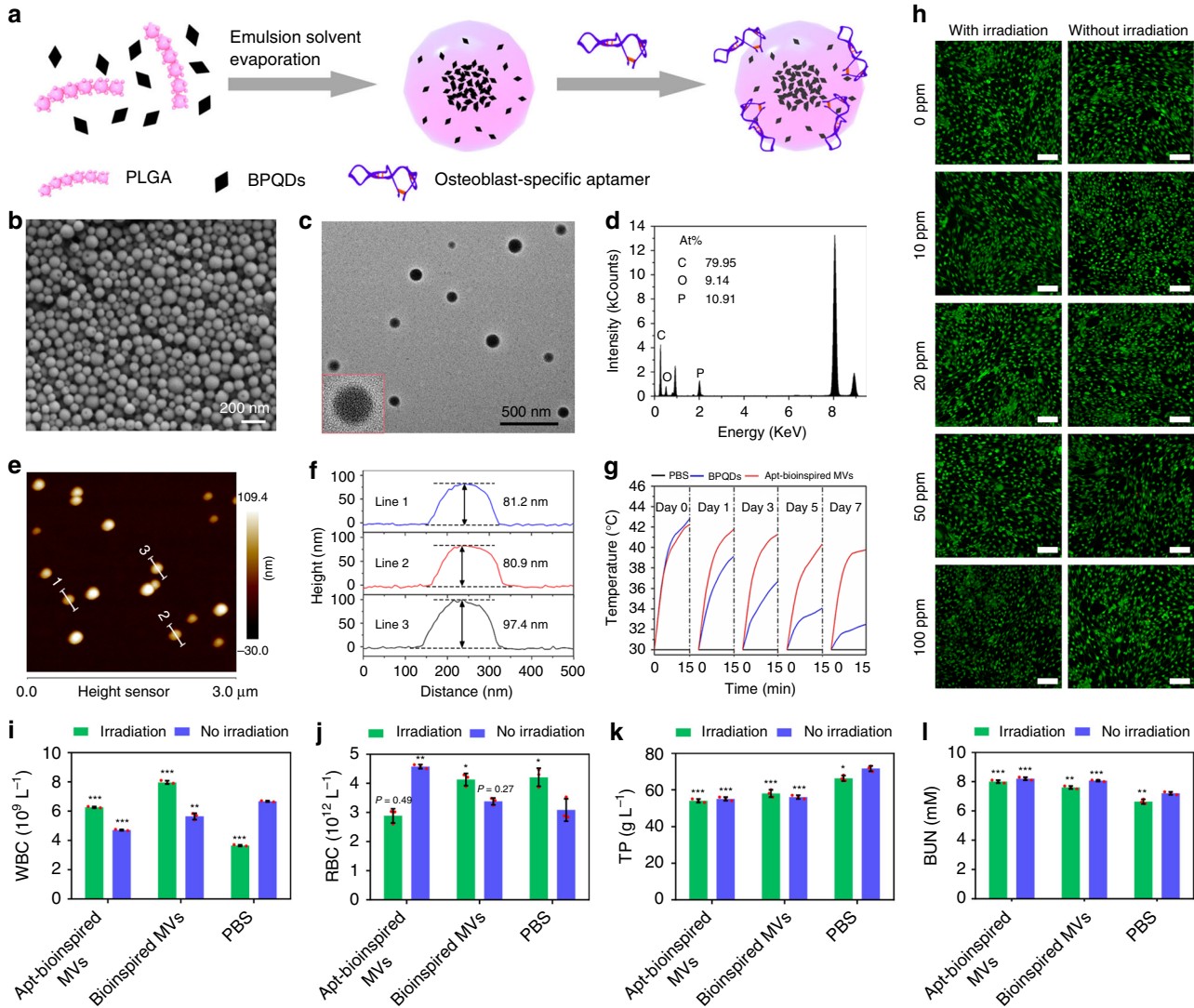

**Fig. 2** Characterization of the Apt-bioinspired MVs. **a** Schematic representation of the preparation of Apt-bioinspired MVs. **b** SEM image, **c** TEM image, **d** EDX analysis, and **e** AFM image of Apt-bioinspired MVs. **f** Height profiles along the white lines in graph **e**. **g** Photothermal curves of bare BPQDs and Apt-bioinspired MVs under NIR irradiation (1.0 W cm$^{-2}$) after storing in PBS for 0, 1, 3, 5, and 7 days. PBS was used as the control. **h** Fluorescence images of dead/live cell staining after incubation with Apt-bioinspired MVs. Scale bars represent 150 μm. Hematological analyses of mice with indexes of (**i**) white blood cell count (WBC) and (**j**) red blood cell count (RBC). Blood biochemical analyses of mice with indexes of (**k**) total protein (TP), and (**l**) blood urea nitrogen (BUN). Data are means ± s.d. ($n = 3$), *$P < 0.05$, **$P < 0.01$, *** $P < 0.001$ (unpaired two-tailed Student's $t$-test). Source data of **b**–**g**, **i**–**l** are provided as a Source Data file

osteoblasts in each sample. As shown in Fig. 4b, without irradiation, the extent of Alizarin Red-stained bone-mineralized nodules in the Apt-bioinspired MVs group is similar to that in the bioinspired MVs group, but is much more than that in the PLGA group. Therefore, it can be concluded that the BPQDs in Apt-bioinspired MVs and the bioinspired MVs have good performance in inducing osteoblast mineralization. The promoted mineralization can be attributed to the upregulated concentration of inorganic phosphates[22,23]. As a further step, the effect of photothermal performance on osteoblast biomineralization was investigated. The level of mineralized nodules in laser-exposed cells containing Apt-bioinspired MVs is almost equal to the level of mineralized nodules in the NIR-irradiated cells containing bioinspired MVs, but is significantly greater than the mineralized nodules in the PLGA group. Importantly, in the Apt-bioinspired MVs group, the extent of osteoblast mineralization is enhanced

by NIR irradiation in both Apt-bioinspired MVs group and bioinspired MVs group. Additionally, the biomineralization capacities of PLGA/phosphates, PLGA/calcium, PLGA/calcium phosphate, and PLGA/osteopontin were also investigated, respectively (Supplementary Fig. 17). The size of the Alizarin red-staining spots in osteoblasts treated with Apt-bioinspired MVs and NIR irradiation is significantly larger than that in the osteoblasts treated with PLGA and phosphates, osteoblasts treated with PLGA and calcium ions, osteoblasts treated with PLGA and calcium phosphate, and osteoblasts treated with osteopontin-embedded PLGA. These results demonstrate that calcium deposition spots in osteoblasts treated with Apt-bioinspired MVs and NIR irradiation are significantly larger than those in the other groups. All of these results clearly suggest that the Apt-bioinspired MVs coupled with NIR irradiation are highly efficient in promoting biomineralization.

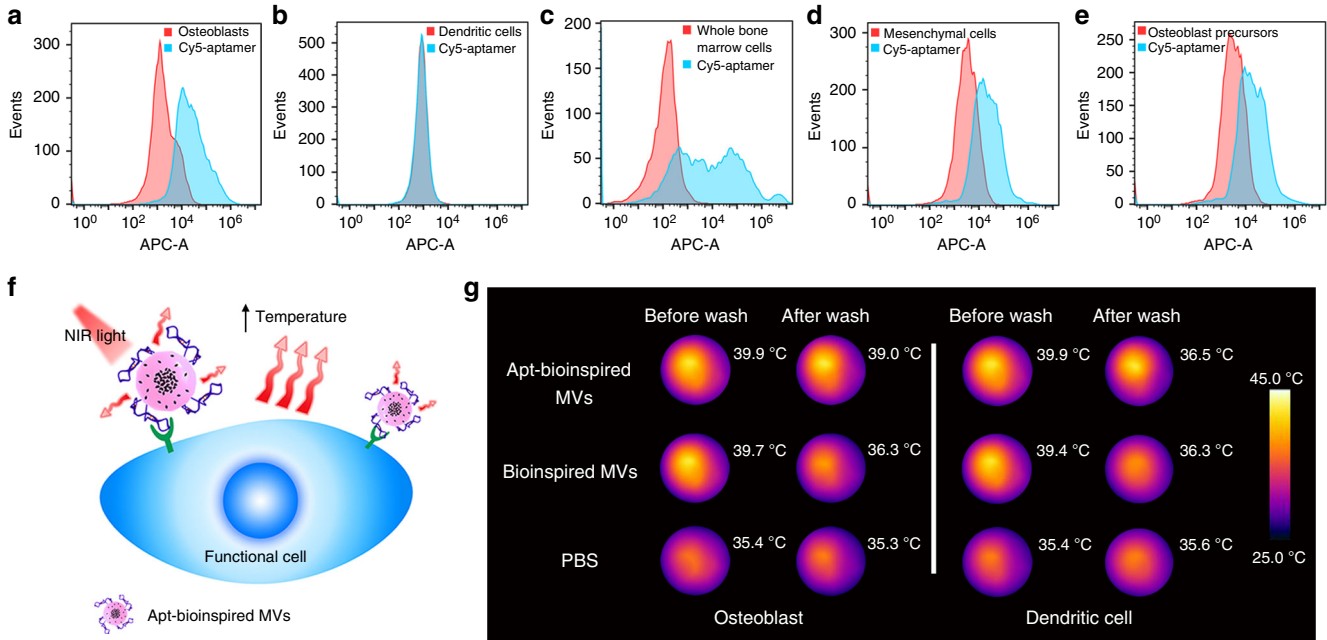

**Fig. 3** In vitro osteoblast targeting ability of Apt-bioinspired MVs. Flow cytometry analysis of **a** rat osteoblasts, **b** rat dendritic cells, **c** whole bone marrow cells, **d** mesenchymal cells, and **e** osteoblast precursor 3T3 cells. All cell populations were counted. **f** Schematic illustration of the binding of Apt-bioinspired MVs with osteoblast through infrared thermal wave imaging based on the photothermal effect. **g** Infrared images of rat osteoblasts and dendritic cells incubated with Apt-bioinspired MVs, bioinspired MVs and PBS before and after washing, respectively. Source data of **a–e** and **g** are provided as a Source Data file

Furthermore, the expression levels of ALP, osteocalcin (OCN), collagen type I alpha 1 (Col1a1), and runt-related transcription factor 2 (Runx2) in osteoblasts were investigated. These proteins are reported to play essential roles in cell mineralization, and some of their expression levels are up-regulated during osteoblastic differentiation and early mineralization-induced osteogenesis[34–36]. Colorimetric assay was employed to study the expression level of ALP. The results show that Apt-bioinspired MVs up-regulate the expression of ALP compared with the other groups (Fig. 4c and Supplementary Fig. 18). Moreover, western blot assays indicate that osteoblasts treated with Apt-bioinspired MVs and NIR irradiation display the highest expression level of Runx2 (Fig. 4d). While the expression levels of OCN and Col1a1 from Apt-bioinspired MVs and NIR irradiation treated osteoblasts are similar to those from untreated osteoblasts (Supplementary Fig. 19). The quantification results of the expression levels of ALP and Runx2 in osteoblasts are presented in Fig. 4e, f, clearly showing the elevated expression of ALP and Runx2 upon treatment with Apt-bioinspired MVs and NIR irradiation. Additionally, the expression level of HSP 70 in osteoblasts treated with Apt-bioinspired MVs with or without NIR irradiation was also determined with western blots analysis (Supplementary Fig. 20). NIR irradiation significantly up-regulated the expression of HSP 70 in Apt-bioinspired MVs cultured osteoblasts, which can be ascribed to the good photothermal performance of the BPQDs in Apt-bioinspired MVs[25–29]. Taken together, the above results confirm that the Apt-bioinspired MVs with photothermal effect can promote biomineralization and the expression of proteins, such as ALP, Runx2, and HSP 70 plays essential roles in the biomineralization process.

**In vivo bone targeting ability of the Apt-bioinspired MVs.** The bone-targeting performance and the clearance of Apt-bioinspired MVs were further investigated by monitoring the in vivo distribution of intravenously injected FITC-loaded Apt-bioinspired

MVs in the main organs and in skull region over time. As shown in Fig. 5, prolonged tracking experiments show that the Apt-bioinspired MVs can reach the skull about 2 h post injection, and large amounts of Apt-bioinspired MVs appears in the skull region after about 2 days of circulation. Moreover, considerable amounts of Apt-bioinspired MVs are still observed in the skull region even after 5 days post injection. Such long time retention of Apt-bioinspired MVs in the skull region is favorable for the reparation of skull bone in subsequent studies. With prolonged time, the amounts of Apt-bioinspired MVs in the main organs and skull region gradually decrease, and much weaker fluorescence signals appear in the organs and skulls, suggesting the gradual clearance of Apt-bioinspired MVs from mice. The above results indicate that the Apt-bioinspired MVs have good capability for targeting the bone region, indicating their ability to promote in vivo biomineralization and bone regeneration.

**In vivo bone defect repair with Apt-bioinspired MVs.** The C57 mice were used as animal models to evaluate the bone defect repair capacity of the Apt-bioinspired MVs. A bone defect with diameter of about 0.5 cm was created on the skull of each C57 mouse. The schematic illustration of bone regeneration is shown in Fig. 6a.

Microcomputed tomography (μ-CT) reconstruction and hematoxylin and eosin (H&E) staining were performed on the bone samples to assess the bone regeneration. The Apt-bioinspired MVs were first intravenously injected into the mice, and good bone regeneration performance was observed (Supplementary Fig. 22). Furthermore, the Apt-bioinspired MVs were directly filled into the bone defects on skulls to investigate the bone defect repair capacity. As shown in Fig. 6b, when the mice are treated with Apt-bioinspired MVs and NIR irradiation, the bone defect is almost completely healed with the generation of large amounts of new bone, both peripherally and centrally. Also, the H&E staining shows that a substantial amount of

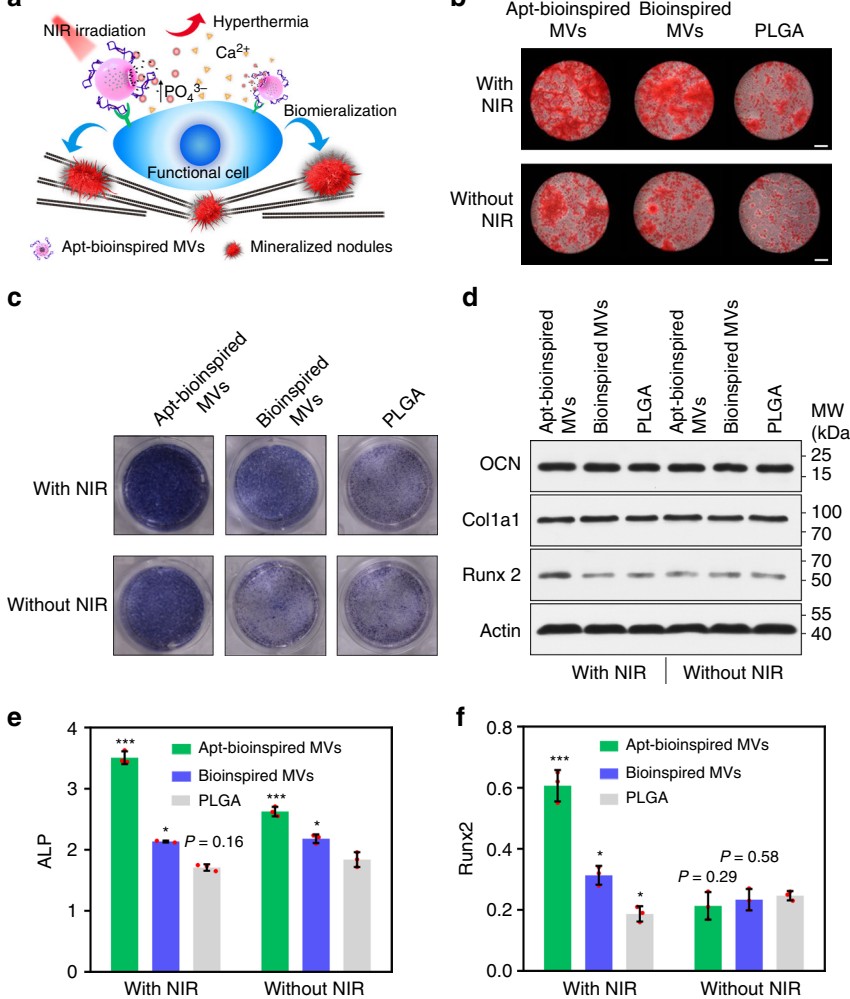

**Fig. 4** In vitro evaluation of Apt-bioinspired MVs-guided biomineralization. **a** Schematic illustration of mechanism of biomineralization. **b** Alizarin Red staining of osteoblasts cultured with Apt-bioinspired MVs (50 ppm), bioinspired MVs (50 ppm), and PLGA NPs (50 ppm) with or without NIR irradiation (0.2 W cm$^{-2}$), respectively. Scale bars are 100 μm. ALP staining analysis (**c**) and western blots analysis of OCN, Col1a1 and Runx2 expression levels (**d**) of osteoblasts treated with Apt-bioinspired MVs (50 ppm), bioinspired MVs (50 ppm), and PLGA NPs (50 ppm) with or without NIR irradiation (0.2 W cm$^{-2}$), respectively. The expression level of (**e**) ALP and (**f**) Runx 2 of osteoblasts cultured with Apt-bioinspired MVs (50 ppm), bioinspired MVs (50 ppm), and PLGA NPs (50 ppm) with or without NIR irradiation (0.2 W cm$^{-2}$), respectively. Data are means ± s.d. ($n = 3$), *$P < 0.05$, **$P < 0.01$, ***$P < 0.001$ (unpaired two-tailed Student's $t$-test). Source data of **d**–**f** are provided as a Source Data file

well-arranged newly formed bone is regenerated, and an increased amount of osteoblastic cells and matrix are exhibited in the defect area (Fig. 6c). However, compared with the NIR-irradiated mice, the amount of newly regenerated bone in mice treated with Apt-bioinspired MVs alone is found to be significantly lower, confirming that the Apt-bioinspired MVs-triggered photothermal effect plays an important role in promoting in vivo biomineralization and bone regeneration (Fig. 6d, e). When the mice are treated with NIR irradiation alone, small amounts of new bone are generated in the defect region (Fig. 6f, g). As for PBS-treated mice, there is no newly formed bone in the bone defect (Fig. 6h, i), indicating that the bone defect cannot be repaired by self-healing. Quantitative analyses including bone defect area and bone volume fraction also demonstrate that Apt-bioinspired MVs-treated group with NIR irradiation exhibits the minimal bone defect area and the maximum bone volume fraction (Fig. 6j, k). Figure 6l presents the concentration variation of phosphate anion over 10 days in the bone defect region of mice treated with Apt-bioinspired MVs

and NIR irradiation. Compared with mice treated with PBS, a significantly higher concentration of phosphate ions is observed in mice treated with Apt-bioinspired MVs and NIR irradiation, suggesting that the Apt-bioinspired MVs can provide phosphate anion for bone regeneration[22,23]. These results verify that Apt-bioinspired MVs exhibit outstanding performance for mineralization-induced bone formation.

In this work, aptamer-guided bioinspired MVs were constructed for stimulating biomineralization and in vivo bone regeneration. The aptamer can direct the bioinspired MVs to the targeted cells in bone region. The important mineral component of inorganic phosphate is obtained from the biodegradable BPQDs. After incubating with osteoblasts, the Apt-bioinspired MVs coupled with the photothermal effect exhibits good performance in accelerating cell differentiation and biomineralization. In addition, the Apt-bioinspired MVs exhibit outstanding capacity in facilitating in vivo biomineralization and further promoting new bone generation. This strategy can inspire researchers to exploit bioinspired materials for regulating

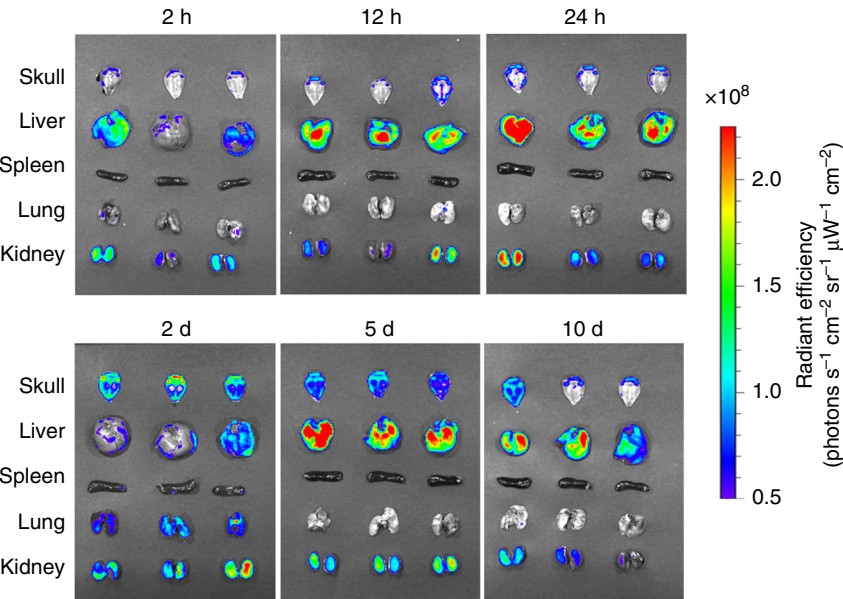

**Fig. 5** In vivo bone-targeting ability of Apt-bioinspired MVs. Ex vivo fluorescence images of isolated skull and main organs from mice injected with FITC-loaded Apt-bioinspired MVs over time. Source data of this figure are provided as a Source Data file

mineralization-guided biological behavior and promote future applications of bioinspired materials in biological and medical engineering.

## Methods

**Materials**. The BP crystals were provided by Nanjing XFNANO Materials Tech Co., Ltd (Nanjing, China). The poly(lactic-co-glycolic acid) (PLGA, AR) (50:50, MW: 30,000−60,000), polyvinyl alcohol (PVA, AR) (MW: 9000−10,000), 1-ethyl-3-(3-dimethylamino-propyl) carbodiimide hydrochloride (EDC, AR), N-hydroxysuccinimide (NHS, AR), β-glycerophosphate, Safranin-O/Fast Green staining, 3′,6′-Di(O-acetyl)-4′,5′-bis[N,N-bis(carboxymethyl) aminomethyl] fluorescein, tetraacetoxymethyl ester (Calcein-AM) and PI were purchased from Sigma-Aldrich. The N-methyl-2-pyrrolidone (NMP), ethanol (AR), sodium hydroxide (NaOH, AR), magnesium chloride hexahydrate ($MgCl_2 \cdot 6H_2O$, AR), paraformaldehyde [$(CH_2O)_n$], AR], ethylene diamine tetraacetic acid (EDTA, analytical grade) and dichloromethane (DCM) were obtained from Sinopharm Chemical Reagent Co., Ltd., China. BCIP/NBT Alkaline Phosphatase Color Development Kit were provided by Beyotime Institute of Biotechnology. The Dulbecco's phosphate buffered saline (D-PBS), α-modified Eagle's medium (α-MEM) and fetal bovine serum (FBS) were obtained from Thermo Fisher Scientific Inc. Ultrapure water was obtained using a Millipore water purification system. 3T3 cells were purchased from Procell Life Science &Technology Co., Ltd. (Catalog #: CP-R162, Wuhan, China). 3T3 cells were purchased from Procell Life Science &Technology Co., Ltd. (Catalog #: CL-0251, Wuhan, China). Osteoblasts, whole bone marrow cell, and mesenchymal cell were isolated from murine bone marrow.

**Aptamers**. Aptamers were all synthesized on an ABI3400 DNA/RNA synthesizer (Applied Biosystems, Foster City, CA, USA). The as-synthesized aptamers were purified by HPLC (Agilent, 1260 GC, Japan) with a C18 column.
   Osteoblast-targeting aptamer:
5′-AGTCTGTTGGACCGAATCCCGTGGACGCACCCTTTGGACG-3′
Cy5-labeled aptamer:
5′-Cy5-AGTCTGTTGGACCGAATCCCGTGGACGCACCCTTTGGACG-3′
Amino-modified aptamer:
5′- $NH_2$-$(A)_9$-AGTCTGTTGGACCGAATCCCGTGGACGCACCCTTTGGACG-3′

**Sample characterization**. The morphologies of BPQDs were recorded with a transmission electron microscope (TEM) with a working voltage of 200 kV (JEOL, JEM-2100, Japan). The TEM imaging of Apt-bioinspired MVs and energy dispersive X-ray analysis were performed using a field-emission TEM (Talos F200S). The SEM image of Apt-bioinspired MVs was obtained on a field-emission scanning electron microscope (Zeiss Merlin Compact). Atomic force microscope (AFM, Veeco, NanoMan) was performed using Bruker Multimode 8 operated in tapping mode. The NIR laser-induced heat conversion curves and photos were obtained with the FLIR A35 infrared thermal imaging camera (USA). UV–vis absorbance spectra were recorded on a UV–vis absorbance spectrometer (UV-2550, Shimadzu,

Japan). The cell viability was tested on a confocal laser scanning microscope (FV1200, Olympus, Japan). The binding of aptamer with cells was evaluated by an Olympus D71 fluorescence microscope (Olympus, Japan). The tissue distribution was evaluated on an IVIS Lumina XR Imaging System (Caliper, America). Microcomputed tomography (μ-CT) reconstruction was recorded by the μ-CT imaging system (μ-CT50, Bruker Skyscan1172, Germany). The histomorphological analysis was carried out on an Olympus DP72 microscope (Olympus, Japan).

**Synthesis of BPQDs**. The BPQDs were prepared using a liquid exfoliation technique[24]. In detail, 200 mg of BP powder was added to 300 mL of NMP. The resulting solution was further subjected to sonication in an ice bath with a sonic tip (ultrasonic frequency: 19−25 kHz) for 8 h (period of 2 s with the interval of 0.1 s) and for another 6 h (period of 2 s with the interval of 4 s) under Ar gas protection. The resultant brown dispersion was centrifuged for 10 min at $3819 \times g$ (Hunan Xiang Yi Laboratory Instrument Development Co., Ltd., TG16-WS, Rotor 7) to remove the un-exfoliated particles and the BPQD supernatant was collected for further use.

**Preparation of bioinspired MVs**. The Bioinspired MVs were prepared by oil-in-water emulsion and solvent evaporation[19]. In particularly, BPQDs in NMP (10 mL) were centrifuged for 20 min at $12,330 \times g$ (Hence Tech Co., Ltd, TG20G, Rotor 2). The obtained precipitate was collected and redispersed in DCM (1 mL) containing PLGA with a concentration of 10 mg mL$^{-1}$. After 15 min of sonication, the mixture was dispersed in 10 mL of 0.5% (w/v) PVA aqueous solution. Then the resultant solution was further sonicated for another 10 min. The obtained emulsion was stirred overnight at room temperature to remove the residual DCM. Lastly, the suspension was centrifuged at $6579 \times g$ (Hunan Xiang Yi Laboratory Instrument Development Co., Ltd, TG16-WS, Rotor 3) for 10 min and washed three times with PB buffer solution (0.01 M, pH = 6.8) to remove free BPQDs, and then redispersed in 1 mL the PB buffer (0.01 M, pH 6.8).

**Preparation of aptamer-functionalized bioinspired MVs**. Typically, 1 mL of Bioinspired MVs in the PB buffer solution (0.01 M, pH = 6.8) was mixed with EDC/NHS (1:1.5) and agitated for 30 min at 125 rpm at 25 °C to activate the carboxyl groups on the surface of bioinspired MVs. The solution was washed twice with PB buffer solution (0.01 M, pH = 7.2) to remove the excess EDC and NHS. The precipitate was redispersed with 2 mL of the PB buffer solution (0.01 M, pH = 7.2). Then, the suspension was reacted with amino-modified aptamer for 4 h by gentle shaking at 125 rpm at 25 °C. The aptamer-functionalized bioinspired MVs (denoted as Apt-bioinspired MVs) were purified by washing with PB buffer solution (0.01 M, pH = 7.2).

**NIR laser-induced heat conversion**. Samples were dispersed in ultrapure water to give concentrations of 20 ppm. The solutions were irradiated with an 808 nm laser (BWT Beijing Ltd., Beijing, China) with a power density of 1 W cm$^{-2}$ for 15 min and the temperature was monitored by the FLIR A35 infrared thermal imaging camera. Ultrapure water was used as the control. NIR illumination at 1 W cm$^{-2}$ for

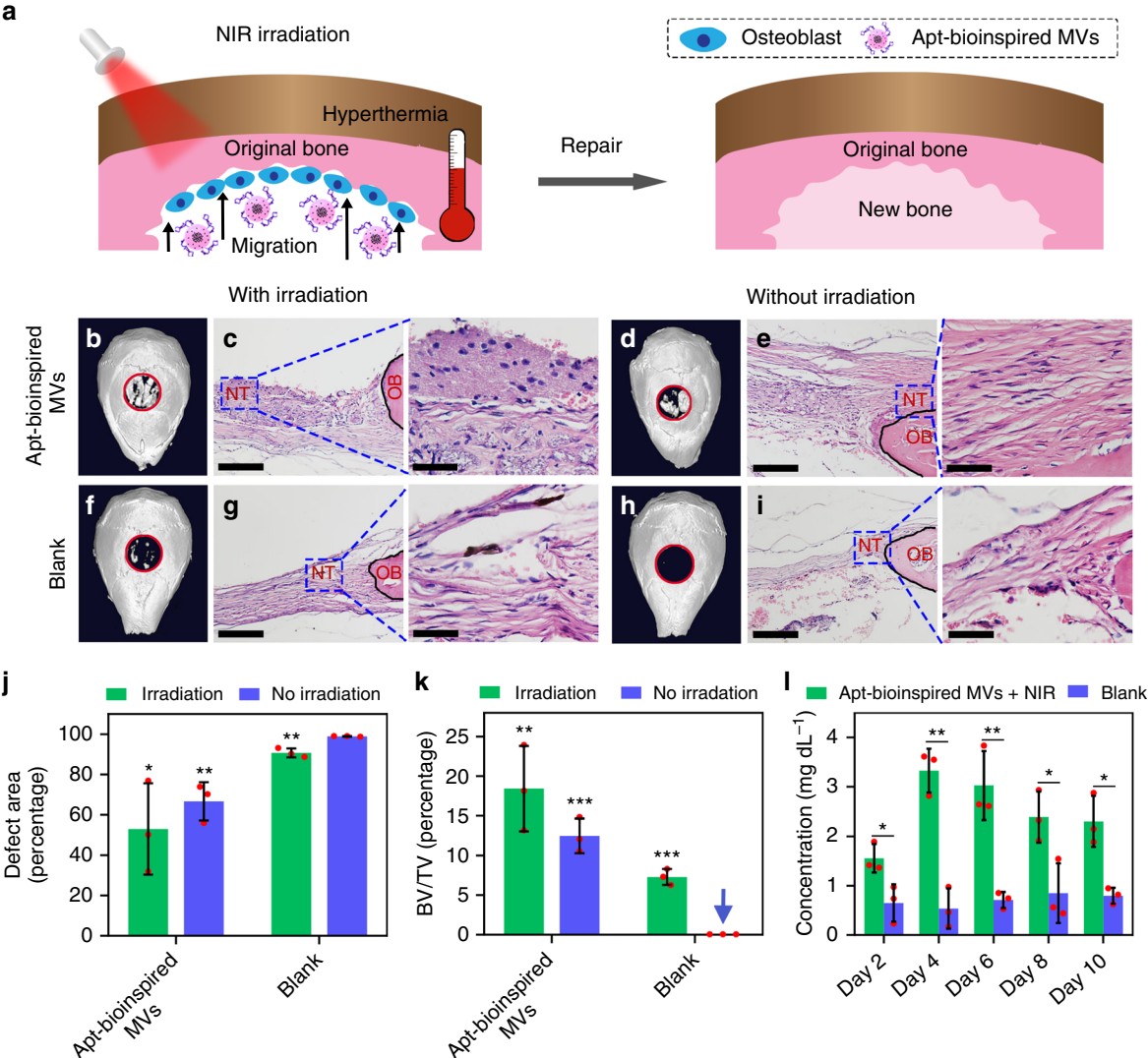

**Fig. 6** In vivo bone defect repair with the Apt-bioinspired MVs. **a** Schematic illustration of bone repair with Apt-bioinspired MVs. **b** μ-CT reconstruction of the bone defect and **c** histomorphological analysis of the tissue after treatment with Apt-bioinspired MVs and NIR irradiation. **d** μ-CT reconstruction of the bone defect and **e** histomorphological analysis of the tissue after treatment with Apt-bioinspired MVs. **f** μ-CT reconstruction of the bone defect and **g** histomorphological analysis of the tissue after treatment with PBS and NIR irradiation. **h** μ-CT reconstruction of the bone defect and **i** histomorphological analysis of the tissue after treatment with PBS. The red rings represent the bone defects. The black solid lines denote the origin bone margins. NT represents newly formed tissue, and OB represents origin bone. **j** Quantification of defect area and **k** quantification of bone volume fraction (BV/TV, bone volume/tissue volume). **l** Concentration variation of phosphate ions over 10 days in the bone defect region of mice treated with Apt-bioinspired MVs under NIR irradiation and PBS, respectively. Scale bars in **c**, **g**, **e**, **f** and the corresponding insets are 100 and 50 μm, respectively. Data are means ± s.d. ($n = 3$), *$P < 0.05$, **$P < 0.01$, ***$P < 0.001$ (unpaired two-tailed Student's $t$-test). Source data of **b–l** are provided as a Source Data file

15 min was used to increase the temperature of the original Apt-bioinspired MVs solution to about 40 °C.

**In vitro cytotoxicity assays**. Typically, $10^5$ rat osteoblasts were incubated with different concentrations of Apt-bioinspired MVs (0, 10, 20, 50, 100 ppm) in culture medium at 37 °C in a 5% $CO_2$ incubator for 4 h. Each sample was divided into two groups. One group was cultured for 4 h without 808 nm laser irradiation. The other group was illuminated with 808 nm laser (0.2 W cm$^{-2}$) for 30 s and then cultured at 37 °C in a 5% $CO_2$ incubator for 4 h. Then, each group of cells was stained with Calcein-AM and PI. The stained cells were further monitored by confocal laser scanning microscopy. Furthermore, the cell viability was quantitatively analyzed by CCK-8 assay. The osteoblasts were incubated with 500 μL of CCK-8 mixture (reagent:culture medium = 1:10) for 4 h and the absorbance at 450 nm was used as the indicator of viable cells.

**Fluorescence microscopy**. Typically, $10^5$ osteoblasts and dendritic cells were incubated with 10 nM Cy5-labeled aptamer in 200 μL of buffer A (adding tRNA and

BSA into buffer B to give concentrations of 0.1 and 1 mg mL$^{-1}$, respectively) for 1 h at 37 °C. After washing with buffer B (adding MgCl$_2$ and glucose into D-PBS with concentrations of 5 mM and 4.5 mg mL$^{-1}$, respectively) for three times, cells were imaged using a fluorescence microscope. Dendritic cells were used as the control.

**Flow cytometry**. Briefly, $10^6$ osteoblasts, dendritic cells, mesenchymal cells, and osteoblast precursor 3T3 cells were incubated with Cy5-labeled aptamer (10 nM) in 200 μL of buffer A (adding tRNA and BSA into buffer B to give concentrations of 0.1 and 1 mg mL$^{-1}$, respectively) for 1 h at 37 °C, respectively. About 5 mL of fresh rat bone marrow was lysed with ammonium chloride. Then the obtained cells were also incubated with Cy5-labeled aptamer (10 nM) in 200 L of buffer A for 1 h at 37 °C. After washing three times with buffer B (adding MgCl$_2$ and glucose into D-PBS with concentrations of 5 mM and 4.5 mg mL$^{-1}$, respectively), the cells were all analyzed by flow cytometry (FACS AriaIII/BD cytometer, America).

**In vitro targeted binding**. Typically, $10^5$ cells were seeded into the pore plates one day before conducting the experiment. Then, Apt-Bioinspired MVs, Bioinspired

MVs were added to the pore plates. The pore plates were placed in the upper chamber and incubated for 4 h. The culture medium in the pore plates was removed, and the pore plates were irradiated with the 808 nm laser (0.2 W cm$^{-2}$) for 30 s. NIR illumination with 0.2 W cm$^{-2}$ for 30 s was used to increase the temperature of the rat osteoblast treated with the Apt-bioinspired MVs to about 40 °C. After that, the temperature of the pore plates was monitored and the IR images were captured using the FLIR A35 infrared thermal imaging camera. Then, the pore plates were washed three times, and the infrared images were further captured. PBS was used as the control.

**In vitro degradation of Apt-bioinspired MVs**. Briefly, 1 mL of Apt-bioinspired MVs (50 ppm) was dispersed in water. The solution was kept in closed sample vials at 37 °C. At certain time intervals, the Apt-bioinspired MVs were centrifuged at 3090×$g$ (Hence Tech Co., Ltd, TG20G, Rotor 2) for 10 min. The supernatant was collected and the concentration of phosphate anions was further analyzed by ion exchange chromatography.

**In vitro biomineralization of rat osteoblasts**. Typically, 10$^5$ osteoblasts were cultured in osteogenic inducing medium (α-MEM containing 10% FBS, 10 nM dexamethasone, 50 µg mL$^{-1}$ ascorbic acid, and 10 mM β-glycerophosphate). The Apt-bioinspired MVs, Bioinspired MVs, and PLGA NPs were added to the plates, respectively. The final concentration of materials was 50 ppm. Each sample was divided into two groups. Typically, one group was cultured for 14 days without the 808 nm laser irradiation. The other group was illuminated with an 808 nm laser (0.2 W cm$^{-2}$) for 30 s every 6 h and cultured for 14 days. The osteogenic inducing medium was changed every 3 days at 37 °C in a 5% CO$_2$ incubator. Next, Alizarin Red S staining, ALP staining and Western blotting were performed to evaluate the mineralization degree of osteoblasts.

**Alizarin Red S staining**. After carefully removing the culture medium, the cells were gently washed with PBS, followed by fixation with 95% ethanol for 10 min. After that, the cells were carefully washed with distilled deionized water (dd H$_2$O) three times. Next, the cells were dark-incubated with 0.1% Alizarin Red S staining solution (pH 4.1) for 30 min. The Alizarin Red S-stained minerals were observed by a microscope after washing with dd H$_2$O.

**ALP staining**. After carefully removing the culture medium, the cells were fixed with paraformaldehyde solution (4%) for 15 min on ice, followed by washing with PBS. After that, the cells were assayed using a BCIP/NBT ALP color development kit (Beyotime Institute of Biotechnology) for 15 min. After washing with dd H$_2$O, the ALP staining levels were recorded by a camera.

**Western blotting**. After carefully removing the culture medium and gently washing with PBS buffer three times, the obtained cells were lysed in RIPA buffer (Servicebio, Inc.) and on ice for 5 min. The cells were collected into a tube, followed by incubating in an ice bath for 30 min. After centrifugation at 12,000 rpm for 10 min, the supernatant containing TP was collected. The samples were denatured at 95 °C for 15 min in 5 × sample buffer (Servicebio, Inc.), separated on 10% SDS–polyacrylamide gels and then transferred onto a polyvinylidene fluoride membrane (Millipore, USA). The membrane was blocked with 5% skim milk in TBS-T buffer (Servicebio, Inc.) for 1 h at room temperature. Then the membrane was further incubated with the primary antibodies against mouse OCN (1:1,000, Catalog # SC-365797, Santa, USA), rabbit Col1a1 (1:1,000, Catalog # bs-7158R, Bioss, China), rabbit Runx 2 (1:1000, Catalog # ab23981, Abcam, UK), rabbit HSP 70 (1:50,000, Catalog # 10995-1-ap, Proteintech Group, Inc., China) and mouse actin (1:3000, Catalog # GB12001, Servicebio, Inc., China), followed by a horseradish peroxidase-conjugated anti-rabbit or anti-mouse IgG. The membranes were enhanced by SuperSignal reagents (Thermo Fisher Scientific).

**Animal handling**. Healthy adult female C57 mice (6 weeks) weighing 17–19 g were purchased from Spfanimals (Beijing, China). The mice were randomly divided into four groups and a circular defect with diameter of 5 mm were made on the skull of each mouse by a trephine. The bone defect repair experiments were conducted by intracranial filling the mice with freeze-dried Apt-bioinspired MVs. The weight of the used Apt-bioinspired MVs was 5.0 mg, and the amount of BPQDs was about 1.2 mg in the Apt-bioinspired MVs. The bone defect mice injected with PBS were used as control. Half of mice groups filled with Apt-bioinspired MVs or PBS were raised for 28 days without NIR laser irradiation. Half of mice groups filled with Apt-bioinspired MVs or PBS were irradiated with the 808 nm laser irradiation (0.2 W cm$^{-2}$) for 10 min every 12 h and were raised for 28 days. NIR illumination with 0.2 W cm$^{-2}$ for 10 min was used to increase the temperature in the defect region of mice treated with Apt-bioinspired MVs to about 40 °C. After the mice were sacrificed at day 28, the skulls were collected surgically and were further immersed in the paraformaldehyde solution (4%) for 24 h at 4 °C. The newly generated bone was monitored by µ-CT-imaging system[37–39]. All studies were performed in compliance with the relevant ethical regulations for animal testing and research. All surgical procedures used in this experiment were approved by the Ethics Committee for Animal Research, Wuhan University, China.

**In vivo degradation of Apt-bioinspired MVs**. Six-week-old female C57 mice were used for the in vivo degradation experiments. A circular defect with diameters of 5 mm was made on the skull of each mouse by a trephine. Then the mice were randomly divided into two groups and the bone defect repair experiments were conducted by intracranial filling with 5 mg of freeze-dried Apt-bioinspired MVs. The bone defect mice injected with PBS were used as control. All mice were irradiated with 808 nm laser irradiation (0.2 W cm$^{-2}$) for 10 min every 12 h. At predetermined time point, the mice were sacrificed. The defect area was rinsed with 60 µL of ddH$_2$O and then 50 µL of the rinsed solution was collected for further analysis. The concentration of phosphate anions was measured by the Phosphate Assay Kit (BioAssay Systems). All studies were performed in compliance with the relevant ethical regulations for animal testing and research. All surgical procedures used in this experiment were approved by the Ethics Committee for Animal Research, Wuhan University, China.

**µ-CT characterization**. Briefly, the skull samples were placed in the position where the long axis of the drilled channel was perpendicular to the axis of the X-ray beam. The X-Ray tube settings were 60 kV and 133 µA and µ-CT images were acquired at 9 µm resolution. A 0.5° rotation step through a 360° angular range with 1.2 s exposure per step was used. The region of the interest (ROI) of 5.0 mm diameter region was defined to assess the extent of bone repair. After 3D reconstruction, the bone defect area and bone volume fractions (BV/TV) were all analyzed by the built-in software of the employed µ-CT.

**Histological and immunohistochemical analyses**. After µ-CT scanning, skull samples were immersed in the EDTA solution (10%) that was refreshed twice weekly. After 4 weeks, the samples were subjected to gradient dehydration for embedding in paraffin. Serial sections of about 7 mm were collected and further fixed on polylysine-coated slides for H&E staining. Observation was performed under a light microscope. Three samples were analyzed for each group.

**Statistical analysis**. The statistical analysis was performed using OriginPro software (version 9.0, OriginLab). The data were expressed as mean ± standard deviation (SD) and all $p$-values were determined by the unpaired $t$-test. A 5% level of significance ($p < 0.05$) was adopted.

## Data availability
The data supporting the findings of this study are available within the article and the Supplementary Information. The source data underlying Figs. 2b–g, i–l, 3a–e, g, 4d–f, 5, 6b–l, and supplementary Figs. 1–7, 9–14, 16, 18, 19, 21, 22m, 22n, and 23 are provided as a Source Data file. All data are available from the corresponding author upon reasonable request.

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

## Acknowledgements

This work was supported by the National Key R&D Program of China (2017YFA0208000, 2016YFF0100800), National Natural Science Foundation of China (21675120), Ten Thousand Talents Program for Young Talents for Prof. Q. Yuan, Fundamental Research Funds for the Central Universities (2042017KF0243). Q. Yuan thanks the large-scale instrument and equipment sharing foundation of Wuhan University.

## Author contributions

Q.Y., Y.F.Z. and J.W. proposed the research direction and guided the project. Y.Q.W., X.X.H. and L.L.Z. designed and conducted the experiments, as well as analyzed the experimental results and drafted the manuscript. C.L.Z. performed the synthesis of the Apt-bioinspired MVs and conducted the photothermal performance tests. Y.X.L. drew the pictures of schematic illustration. Y.L.W. and C.W. performed in vivo animal experiments. All the authors checked the manuscript.

## Additional information

**Competing interests:** The authors declare no competing interests.

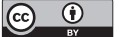

