## [Peer Review File · Nature Communications]

Reviewers' comments:

Reviewer #1 (Remarks to the Author):

The authors are realized a very good work but regarding the remineralization I am not convinced (Fig.6). From the images, the bone formation was minimal, above the untreated groups, perhaps more due to the systemic injection than the evaluation time. Shape bone, but too little to justify proposing such treatment.

The quality of the work guet lost with the conclusion and we suggest more experiments to confirm the complete bone formation to strengthen the conclusions.

Reviewer #2 (Remarks to the Author):

The manuscript is very interesting in new innovative method for bone repair. Although the physical analysis in new variable bio-medicine targeted osteoblasts is very good approach there are some basal serious problems. There are the many concerns need to be carefully addressed.

<Major comments>

1) Although the authors performed the Apt-bioinspired MVs with only phosphorus, osteoblasts have well known be deposited calcium ions as well as phosphate during bone mineralization. Why didn't the authors examine using Apt-bioinspired MVs embedded with the calcium and/or phosphate in the present experiments? The authors should compare the Apt-bioinspired MVs with phosphates to that of calcium and calcium phosphates.

Furthermore, the bone repair need the organic component such as collagen, osteoblast secreted proteins (osteopontin, osteocalcin, and BP). The phosphorus is a component in many bone matrixes. If possible, the authors should be tried to check the MV including some osteogenesis proteins in present experiments.

2) The authors stated that BPQDs in humid environment can degradate into inorganic phosphates. Therefore, the authors should measure actual concentration of degradated phosphates at least in vitro present experiments.

3) There are some questions to be address; i) how did the Apt-bioinspired MVs reach to the bone defect site from intravenous injection? Usually, the MVs quickly are excreted from kidney few days. Did the authors check how long the MVs disappear in each organ in the time course? ii) Why did the irradiation with NIR upregulate the mineral deposition? What type of proteins, such as heat shock proteins and/or phosphate transporters (pit1 and pit2) did the NIR irradiation activate in the osteoblasts?

The authors should examine the additional experiments and submit supplemental data to clarify the questions.

4) The osteoblasts recruit into bone repair area and differentiated into mature osteoblasts from osteoblast precursors and mesenchymal cells. Therefore, are the aptamer to the cells should maintain to bind to all of osteoblast ancestor cells. Did the authors check their duration of binding affinity?

<Minor comments>

1) Fig. 4: The authors should examine the expression of osteogenesis related proteins because the amount of mRNAs changes to that of the proteins after post-transcriptional modification.

2) It is confused in the similar words between biomimetic and bioinspired.

3) It is unclear the difference between extracellular vesicles and matrix vesicles.

Responses to the Referees' Comments and the Corresponding Revisions

We thank the referees for their positive endorsement and pertinent comments about our paper. We have carefully considered the referees' comments and tried to modify the manuscript accordingly. Our responses and corresponding revisions are as follows:

Response to Reviewer 1:

Comment: The authors are realized a very good work but regarding the remineralization I am not convinced (Fig.6). From the images, the bone formation was minimal, above the untreated groups, perhaps more due to the systemic injection than the evaluation time. Shape bone, but too little to justify proposing such treatment. The quality of the work guet lost with the conclusion and we suggest more experiments to confirm the complete bone formation to strengthen the conclusions.

Response: Thanks very much to the reviewer for his/her valuable suggestion. We have conducted bone defect repair experiments by intracranial filling mice with different kinds of MVs, and nearly complete bone formation was observed in mice treated with Apt-bioinspired MVs under NIR irradiation.

The microcomputed tomography (μ -CT) image in Figure 1a below shows that mice treated with the Apt-bioinspired MVs and NIR irradiation display nearly complete bone regeneration. The histomor and phological (H&E) staining analysis further confirms the generation of well-arranged new bone (Figure 1b below). Whereas mice received other treatments show partially bone formation (Figure 1c-j below). These results suggest that the Apt-bioinspired MVs together with NIR irradiation can efficiently promote bone defect repair. Figure 1 below was added into the revised manuscript as Figure 6.

Figure 1. (a) μ -CT image of the skull and (b) histomor-phological analysis of the skull tissue from the mice treated with Apt-bioinspired MVs and NIR irradiation. (c) μ -CT image of the skull and (d) histomor-phological analysis of the skull tissue from the mice treated with Apt-bioinspired MVs. (e) μ -CT image of the skull and (f) histomor-phological analysis of the skull tissue from the mice

treated with PBS and NIR irradiation. (g) μ -CT image of the skull and (h) histomorphological analysis of the skull tissue from the mice treated with PBS. The red circles show the bone defects. The black solid lines denote the original bone margins. NT represents newly formed tissue, and OB represents original bone. (i) Quantification of defect area and (j) quantification of bone volume density (BV/TV, bone volume/tissue volume). Scale bars in (c), (g), (e), (f) and the corresponding insets are 100 μ m and 50 μ m, respectively.

Response to Reviewer 2:

The manuscript is very interesting in new innovative method for bone repair. Although the physical analysis in new variable bio-medicine targeted osteoblasts is very good approach there are some basal serious problems. There are the many concerns need to be carefully addressed.

Comment: Although the authors performed the Apt-bioinspired MVs with only phosphorus, osteoblasts have well known be deposited calcium ions as well as phosphate during bone mineralization. Why didn't the authors examine using Apt-bioinspired MVs embedded with the calcium and/or phosphate in the present experiments? The authors should compare the Apt-bioinspired MVs with phosphates to that of calcium and calcium phosphates. Furthermore, the bone repair need the organic component such as collagen, osteoblast secreted proteins (osteopontin, osteocalcin, and BP). The phosphorus is a component in many bone matrixes. If possible, the authors should be tried to check the MV including some osteogenesis proteins in present experiments.

Response: Thanks a lot for the reviewer's valuable suggestion. We have investigated the biomineralization capabilities of MVs embedded different kinds of compounds, and the results showed that the Apt-bioinspired MVs coupled with NIR irradiation displayed the best biomineralization capability, as shown in Figure 2 below. The size of the Alizarin red staining spots in osteoblasts treated with Apt-bioinspired MVs and NIR irradiation are significantly larger than the osteoblasts treated with PLGA, osteoblasts treated with PLGA and phosphates, osteoblasts treated with PLGA and calcium ions, osteoblasts treated with PLGA and calcium phosphate, and osteoblasts treated with osteopontin embedded PLGA. These results illustrate that calcium deposition spots in osteoblasts treated with Apt-bioinspired MVs and NIR irradiation are significantly larger than in other groups. The description about Figure 2 below was added into the revised manuscript, and Figure 2 below was added into the supporting information of the revised manuscript.

Figure 2. Alizarin Red staining of osteoblasts treated with PLGA, osteoblasts treated with PLGA and phosphates, osteoblasts treated with PLGA and calcium ions, osteoblasts treated with PLGA and calcium phosphate, osteoblasts treated with osteopontin embedded PLGA, and osteoblasts treated with Apt-bioinspired MVs and NIR irradiation.

Comment: The authors stated that BPQDs in humid environment can degrade into inorganic phosphates. Therefore, the authors should measure actual concentration of degraded phosphates at least in vitro present experiments.

Response: Thanks very much for the reviewer's kind suggestion. The actual concentration of degraded phosphates was measured, as shown in Figure 3 below. The description about Figure 3 below was added into the revised manuscript, and Figure 3 below was added into the supporting information of the revised manuscript.

Figure 3. The concentration of phosphates released by the Apt-bioinspired MVs versus time.

Comment: There are some questions to be address; i) how did the Apt-bioinspired MVs reach to the bone defect site from intravenous injection? Usually, the MVs quickly are excreted from kidney few days. Did the authors check how long the MVs disappear in each organ in the time course? ii) Why did the irradiation with NIR upregulate the mineral deposition? What type of

proteins, such as heat shock proteins and/or phosphate transporters (pit1 and pit2) did the NIR irradiation activate in the osteoblasts? The authors should examine the additional experiments and submit supplemental data to clarify the questions.

Response: Thanks a lot for the reviewer's kind suggestion.

i) The intravenously injected Apt-bioinspired MVs can reach the bone defect site via the permeable vessels. Previous studies reported that vasculature is the conduit for nutrient exchange between bone and the rest of the body, and new vessels are involved in bone regeneration. Moreover, the bone defect can induce inflammation responses and inflammation responses can increase vascular permeability. These previous studies suggest that the intravenously injected Apt-bioinspired MVs can reach the bone defect site due to the increased vascular permeability. (Towler, D. A. *et al. Nat. Rev. Endocrinol.* **2012**, 8, 529; Cowin, S. C. *et al. Journal of Biomechanics* **2015**, 48, 842; Tabata, Y. *et al. Biomaterials* **2012**, 33, 304; Pober, J. S. *et al. Nat. Rev. Immunol.* **2007**, 7, 803; Ferrari, M. *et al. Nat. Biotechnol.* **2015**, 33, 941.)

According to the reviewer's kind suggestion, we have investigated the distribution of the injected Apt-bioinspired MVs in mice over time. Prolonged tracing experiments show that the Apt-bioinspired MVs are cleared from mice after 10 days post injection, whereas considerable amounts of Apt-bioinspired MVs can reach the bone region within 2 days. Figure 4 below was added into the revised manuscript as Figure 5.

ii) Experiment results show that the level of heat shock proteins 70 (HSP 70) and alkaline phosphatase (ALP) increases in osteoblast treated with the Apt-bioinspired MVs and NIR irradiation, as shown in Figure 5 below. Previous studies reported that hyperthermia can stimulate the up-regulated expression of HST and ALP in biomineralization, suggesting that the photothermal effect of the black phosphorus in the Apt-bioinspired MVs lead to the increased expression of HST 70 and ALP in osteoblast (Scutt, A. *et al. J. Bone Miner. Res.* **2001**, 16, 731; Rattan, S. I. S. *et al. Ann. N. Y. Acad. Sci.* **2006**, 1067, 443; Nishida, Y. *et al. PLoS One* **2017**, 12, e0181404; Kajiya, H. *et al. J. Biomater. Appl.* **2015**, 29, 1109). Figure 5 below was added into the revised manuscript.

Figure 4. *Ex vivo* fluorescence images of isolated skull and main organs from

mice injected with FITC-loaded Apt-bioinspired MVs over time.

Figure 5. (a) Western blots analysis of HSP 70 expression levels of osteoblasts cultured with Apt-bioinspired MVs (50 ppm) with or without NIR irradiation (0.2 W cm^{-2}), respectively. (b) ALP staining analysis of osteoblasts treated with Apt-bioinspired MVs (50 ppm) with or without NIR irradiation (0.2 W cm^{-2}), respectively.

Comment: The osteoblasts recruit into bone repair area and differentiated into mature osteoblasts from osteoblast precursors and mesenchymal cells. Therefore, are the aptamer to the cells should maintain to bind to all of osteoblast ancestor cells. Did the authors check their duration of binding affinity?

Response: Thanks very much to the reviewer.

(i) The aptamer can bind to both osteoblast precursors 3T3 cells and mesenchymal cells, as shown in Figure 6 below. Figure 6 below was added into the revised manuscript as Figure 3d and 3e.

(ii) We have tried to check the duration of binding affinity by measuring the fluorescence intensity of the aptamer treated osteoblasts over time with flow cytometry. The obtained results show irregular changes of fluorescence intensity. We have consulted Prof. Weihong Tan group at Hunan University about the irregular flow cytometry results. They told us that the proliferation of cells can seriously affect the measuring, and it is impossible to get reliable results about the duration of binding affinity.

Figure 6. Flow cytometry analysis of (a) mesenchymal cells and (b) osteoblast precursors 3T3 cells after incubation with Cy5-labeled aptamer.

Comment: Fig. 4: The authors should examine the expression of osteogenesis related proteins because the amount of mRNAs changes to that of the proteins after

post-transcriptional modification.

Response: Thanks a lot to the reviewer for his/her valuable suggestion. The expression level of ALP, osteocalcin (OCN), collagen type I alpha 1 (Col1a1) and runt related transcription factor 2 (Runx2) were examined. The results show that the expression levels of ALP and Runx2 are increased in osteoblasts treated with Apt-bioinspired MVs and NIR irradiation, whereas no obvious increased expression of Col1a1 and OCN is observed (Figure 7 below). These results indicate that the ALP and Runx2 are involved in the biomineralization of osteoblasts treated with Apt-bioinspired MVs and NIR irradiation. Figure 7 below was added into the revised manuscript as Figure 4.

Figure 7. ALP staining analysis (a) and western blots analysis of Runx2, Col 1a1 and OCN expression levels (b) of osteoblasts treated with Apt-bioinspired MVs (50 ppm), bioinspired MVs (50 ppm) and PLGA NPs (50 ppm) with or without NIR irradiation (0.2 W cm^{-2}), respectively. The corresponding quantitative analysis of ALP (c) and Runx2 (d) of osteoblasts treated with Apt-bioinspired MVs (50 ppm), bioinspired MVs (50 ppm) and PLGA NPs (50 ppm) with or without NIR irradiation (0.2 W cm^{-2}), respectively.

Comment: It is confused in the similar words between biomimetic and bioinspired.

Response: Thanks very much to the reviewer. The word “biomimetic” was changed as “bioinspired” throughout the revised manuscript for clarity.

Comment: It is unclear the difference between extracellular vesicles and matrix vesicles.

Response: Thanks a lot to the reviewer. The matrix vesicles (MVs) are a special kind of extracellular vesicles (EVs) released by bone-related functional cells and this information is added into the abstract of the revised manuscript. The EVs are membrane-contained vesicles released in an evolutionally conserved manner by cells. The osteoblasts can secrete a special kind of EVs, known as MVs. The MVs are a special kind of EVs. (Yáñez-Mó, M. *et al. J. Extracell. Vesicles* **2015**, *4*, 27066; Anderson, H. C. *et al. Curr. Rheumatol. Rep.* **2003**, *5*, 222; Shapiro, I. M. *et al. Bone* **2015**, *79*, 29.)

Reviewers' comments:

Reviewer #1 (Remarks to the Author):

The authors are realized a very good work in the new version of the manuscript with the additional information and new figures more completed. The answers of the comments are good and many comments were added in the new version.

So I agree with the authors that the remineralization by bioinspired MVs embedded with black phosphorus display outstanding bone regeneration performance.

Reviewer #2 (Remarks to the Author):

The revised manuscript almost changed as point-out.
However, I am afraid that there are still points to revise.

1) In Figure 2i-l (bar graphs) and Figure 4e and 4f (bar graphs), there is no statistic analysis. The authors should analyze the statistic difference in these experimental conditions. And the authors should add the statistical method in text.

2) Figure 4C: It is unclear that ALP staining using Apt bioinspired MVs had difference between the presence and absence of NIR. The authors quantitatively should analyze the staining images using colorimetric method or absorbance method. Furthermore, the authors quantitatively should represent the ratio of targeted molecules to actin in the WB band images using image J etc.

Reviewer #3 (Remarks to the Author):

In this work, the authors developed aptamer-guided bioinspired matrix vesicles (Apt-bioinspired MVs) based on two-dimensional black phosphorus (BP) for stimulating biomineralization and in vivo bone regeneration. In general, the major concern on the manuscript at current stage is that the novelty of this work is NOT enough to be published in Nature Communications, while BP/PLGA based vesicles and BP used for osteogenesis have been published in other works. In addition, more characterization data needs to be collected while current data can't support the conclusions very well. Therefore, my recommendation is to reject.

Additional comments:

1. BP/PLGA based microspheres used for bone regeneration, the degradation of black phosphorus into phosphate ion used for osteogenesis, and the photothermal effect of BP under NIR laser irradiation used for heat-induced osteogenesis all have been published in other papers (such as Biomaterials, 2018, 179: 164-174; Adv. Mater. 2018, 30, 1705611; Biomaterials, 2019, 193: 1-11), but the author did not cite these papers. Therefore, the novelty of this work is limited.
2. More characterization data of Apt-bioinspired MVs need to be provided, such as the loading efficiency, encapsulation efficiency and the content of each component in the MVs.
3. In the in vivo experiments, the dose of injection is unknown.
4. The power density and irradiation time of NIR laser used in this paper are very confusing (1 W cm⁻² for 15 min in the heat conversion experiments; 0.2 W cm⁻² for 30 s in the in vitro experiments; and 0.2 W cm⁻² for 10 min in the in vivo experiments), please explain it.
5. In Fig. S16, the original concentration of Apt-bioinspired MVs is not clear and detailed experimental procedures need to be provided.
6. More data should be provided to prove that such tiny amount and slow degradation rate of BPQDs in vivo can effectively increase the concentration of inorganic phosphate. Fig. S16 is not enough, since body fluids are dynamic in the body. Furthermore, what is the amount of BPQDs used in vivo

experiments? I think the concentration of PO₄³⁻ in normal body fluids maybe much higher than that degraded from BPQDs.

Responses to the Referees' Comments and the Corresponding Revisions

We thank the referees for their positive endorsement and pertinent comments about our paper. We have carefully considered the referees' comments and tried to modify the manuscript accordingly. Our responses and corresponding revisions are as follows:

Response to Reviewer 1:

Comment: The authors are realized a very good work in the new version of the manuscript with the additional information and new figures more completed. The answers of the comments are good and many comments were added in the new version.

So I agree with the authors that the remineralization by bioinspired MVs embedded with black phosphorus display outstanding bone regeneration performance.

Response: Thanks very much for the reviewer's comments.

Response to Reviewer 2:

The revised manuscript almost changed as point-out. However, I am afraid that there are still points to revise.

Comment: In Figure 2i-l (bar graphs) and Figure 4e and 4f (bar graphs), there is no statistic analysis. The authors should analyze the statistic difference in these experimental conditions. And the authors should add the statistical method in text.

Response: Thanks a lot for the reviewer's valuable suggestion. The statistic difference in these experimental conditions and the statistical methods were added into the revised manuscript.

Comment: Figure 4C: It is unclear that ALP staining using Apt bioinspired MVs had difference between the presence and absence of NIR. The authors quantitatively should analyze the staining images using colorimetric method

or absorbance method. Furthermore, the authors quantitatively should represent the ratio of targeted molecules to actin in the WB band images using image J etc.

Response: Thanks very much for the reviewer's kind suggestion. The ALP staining was quantitatively analyzed and the up-regulated expression of ALP was observed in osteoblasts treated with Apt-bioinspired MVs (Figure 1 below). The ratio of OCN and Col1a1 to actin in the WB band images was shown in Figure 2 below. The ratio of Runx2 to actin has been shown in Figure 4f in the original manuscript. Figures 1 and 2 below were added into the supporting information of the revised manuscript.

Figure 1. The ALP staining analysis of the expression levels of ALP in osteoblasts treated with Apt-bioinspired MVs, bioinspired MVs and PLGA NPs (50 ppm) with or without NIR irradiation, respectively.

Figure 2. The ratio of OCN and Col1a1 to actin in WB band images.

Response to Reviewer 3:

In this work, the authors developed aptamer-guided bioinspired matrix vesicles (Apt-bioinspired MVs) based on two-dimensional black phosphorus (BP) for stimulating biomineralization and in vivo bone regeneration. In general, the major concern on the manuscript at current stage is that the novelty of this work is NOT enough to be published in Nature Communications, while BP/PLGA based vesicles and BP used for osteogenesis have been published in other works. In addition, more characterization data needs to be collected while current data can't support the conclusions very well. Therefore, my recommendation is to reject.

Comment: BP/PLGA based microspheres used for bone regeneration, the degradation of black phosphorus into phosphate ion used for osteogenesis, and the photothermal effect of BP under NIR laser irradiation used for heat-induced osteogenesis all have been published in other papers (such as *Biomaterials*, 2018, 179: 164-174; *Adv. Mater.* 2018, 30, 1705611; *Biomaterials*, 2019, 193: 1-11), but the author did not cite these papers. Therefore, the novelty of this work is limited.

Response: Thanks a lot for the reviewer's comment. We carefully read the three papers mentioned by the reviewer and found that our manuscript is significantly different from these three works. The three papers mentioned by the reviewer were cited in the revised manuscript as ref. 17, 20 and 24.

1) **The paper on *Biomaterials* (2018, 179, 164-174) reported the usage of heat generated by BPQDs to open the PLGA shell for controlled release of SrCl₂.** This paper reported that Sr could promote bone regeneration, whereas the controlled release of Sr from Sr-enriched biomaterials was difficult. Thus, the authors prepared BP-SrCl₂/PLGA microspheres and used the heat generated by BPQDs to open the PLGA shell for controlled release of SrCl₂, as shown in Figure 3 below. **This paper did not report the promotion of bone regeneration by thermal stimulation, and this paper did not report the promotion of bone regeneration by the phosphate ions originated from the BPQDs.** Moreover, the BP-SrCl₂/PLGA microspheres were placed in the defect region by surgical implantation because the size of the BP-SrCl₂/PLGA microspheres (about 45 μm) was much too large for intravenous injection. **In**

our manuscript, we constructed a synergistic system for biomineralization and bone regeneration. That is, aptamer can direct the Apt-bioinspired MVs to targeted cells, the phosphate ions originated from the BPQDs can facilitate cell biomineralization, the heat generated by the Apt-bioinspired MVs can promote the bone regeneration by stimulating the up-regulated expression of heat shock proteins and alkaline phosphatase. In addition, due to the small size and the targeted recognition function of the Apt-bioinspired MVs, the Apt-bioinspired MVs can be used for bone repair either by surgical implantation or by intravenous injection. Therefore, our manuscript is significantly different from this paper on Biomaterials.

- 2) The paper on Advanced Materials (*Adv. Mater.* **2018**, *30*, 1705611) reported the preparation of BP nanosheets embedded bioglass (BG) scaffold for osteosarcoma thermal therapy and bone regeneration. **The BP nanosheets were used to generate heat to kill the osteosarcoma. This paper did not report the promotion of bone regeneration by thermal stimulation**, as shown in Figure 4 below. Also, the bulk scaffold was placed in the defect region by surgical implantation. **In our manuscript, we constructed a synergistic system for biomineralization and bone regeneration. In addition to the phosphate ions originated from the BPQDs, the heat generated by BPQDs can also promote biomineralization.** Moreover, the aptamer can direct the Apt-bioinspired MVs to defect region by recognizing the osteoblasts. The Apt-bioinspired MVs can be used for bone repair either by surgical implantation or by intravenous injection.
- 3) The paper on Biomaterials (*Biomaterials* **2019**, *193*, 1-11) reported the preparation of BPs@PLGA membranes for bone regeneration. **The authors ascribed the enhancement of osteogenesis to the heat generated by BP nanosheet**, as shown in Figure 5 below. The bulk BPs@PLGA membranes were placed in the defect region by surgical implantation. **In our manuscript, we showed that in addition to the heat generated by BPQDs, DNA aptamer and the phosphate ions originated from the BPQDs both could facilitate cell biomineralization and bone regeneration.** Moreover the Apt-bioinspired MVs can be used for bone repair either by surgical implantation or by intravenous injection.

Figure 3. The main design of the paper on Biomaterials (*Biomaterials* 2018, 179, 164-174).

Figure 4. The main design of paper on Advanced Materials (*Adv. Mater.* 2018, 30, 1705611).

Figure 5. The main design of paper on Biomaterials (*Biomaterials*, 2019, 193, 1-11).

Comment: More characterization data of Apt-bioinspired MVs need to be provided, such as the loading efficiency, encapsulation efficiency and the content of each component in the MVs.

Response: Thanks very much to the reviewer for his/her kind suggestion. The loading efficiency of BPQDs in the MVs was determined to be 23.4% according to the results of EDX analysis. The encapsulation efficiency of BPQDs and aptamers by the MVs were 95.7% and 16.0%, respectively. These data were added into the revised manuscript.

Comment: In the *in vivo* experiments, the dose of injection is unknown.

Response: Thanks a lot to the reviewer for his/her kind suggestion. The dose of injection in the *in vivo* experiments was added into the revised manuscript.

Comment: The power density and irradiation time of NIR laser used in this paper are very confusing (1 W cm⁻² for 15 min in the heat conversion experiments; 0.2 W cm⁻² for 30 s in the *in vitro* experiments; and 0.2 W cm⁻² for 10 min in the *in vivo* experiments), please explain it.

Response: Thanks very much for the reviewer's comment. Previous studies showed that temperature about 40 °C is favorable for biomineralization process. So we changed the irradiation parameters to maintain the temperature at about 40 °C in different experiments. (*Tissue Eng., Part A* **2012**, *19*, 716; *Cell Biochem. Funct.* **2007**, *25*, 267)

In the heat conversion experiment, NIR illumination with 1 W cm⁻² for 15 min was used to increase the temperature of the Apt-bioinspired MVs solution to about 40 °C, as shown in Figure 2g in the original manuscript. In the *in vitro* experiments, NIR illumination with 0.2 W cm⁻² for 30 s was used to increase the temperature of the rat osteoblast treated with the Apt-bioinspired MVs to about 40 °C, as shown in Figure 3g in the original manuscript. In the *in vivo* experiment, NIR illumination with 0.2 W cm⁻² for 10 min was used to increase the temperature about the defect region of mice treated with Apt-bioinspired MVs to about 40 °C, as shown in Figure 6 below. Figure 6 below was added into the supporting information of the revised manuscript.

Figure 6. (a) Infrared thermographic maps with the notations of the temperature in the NIR light (0.2 W cm^{-2} for 10 min) irradiated defect region on mice treated with Apt-bioinspired MVs and PBS, respectively. (b) Time-dependent temperature increase in the NIR light (0.2 W cm^{-2} for 10 min) irradiated defect region on mice treated with Apt-bioinspired MVs and PBS, respectively.

Comment: In Fig. S16, the original concentration of Apt-bioinspired MVs is not clear and detailed experimental procedures need to be provided.

Response: Thanks very much for the reviewer's kind suggestion. The original concentration of Apt-bioinspired MVs in Fig. S16 and the detailed experimental procedures about Fig. S16 were added into the revised manuscript.

Comment: More data should be provided to prove that such tiny amount and slow degradation rate of BPQDs in vivo can effectively increase the concentration of inorganic phosphate. Fig. S16 is not enough, since body fluids are dynamic in the body. Furthermore, what is the amount of BPQDs used in vivo experiments? I think the concentration of PO_4^{3-} in normal body fluids may be much higher than that degraded from BPQDs.

Response: Thanks a lot to the reviewer. According to the reviewer's valuable suggestion, we measured the concentration variation of phosphate anion in the bone defect region over 10 days. The concentrations of phosphate anion in the bone defect region of mice treated with the Apt-bioinspired MVs and NIR light were higher than that in the bone defect region of mice treated with PBS, as shown in Figure 7 below. Figure 7 below and the corresponding description were added into the revised manuscript.

The amount of the added Apt-bioinspired MVs was 5.0 mg in the *in vivo* experiments. Considering that the loading efficiency of BPQDs in PLGA nanoparticles was 23.4%, the amount of BPQDs was determined to be about 1.2 mg.

Figure 7. The concentration variation of phosphate anion in the bone defect region of mice treated with the Apt-bioinspired MVs/NIR or PBS over 10 days. Data are means \pm SD (n=3), *P<0.05, **P<0.01 (unpaired t-test).

Reviewers' comments:

Reviewer #1 (Remarks to the Author):

I confirm my previous comments: The authors are realized a very good work in the new version of the manuscript with the additional information and new figures more completed. The answers of the comments are good and many comments were added in this new version.

Reviewer #2 (Remarks to the Author):

The authors have revised the manuscript as my pointed-out portions. So I feel like to be enough the manuscript for acceptance in your journal.

Reviewer #3 (Remarks to the Author):

1. As I have mentioned before, More data should be provided to prove that such tiny amount and slow degradation rate of BPQDs in vivo can effective increase the concentration of inorganic phosphate. Although the authors have added an experiment, I still think that the concentration of PO_4^{3-} in normal body fluids is very much higher than that degraded from BPQDs. What amount of the BP was used? Even if the increased PO_4^{3-} is just from BP, why such short time NIR light irradiation in vivo can cause the degradation of PLGA and BP?
2. The authors want to show a strategy for Molecular Recognition-Guided Biomineralization. As indicated in Figure 1, a tail vein injection was proposed. And the authors mentioned "cell-targeting aptamer modified bioinspired MVs (denoted as Apt-bioinspired MVs) are designed for promoting cell biomineralization (Fig. 1). The aptamer can guide the bioinspired MVs to targeted cells." However, the authors choose intracranial filling in the followed animal experiments for bone defect repair. I think tail vein injection of the Apt-bioinspired MVs is required for the demonstration of the Molecular Recognition-Guided Biomineralization.
3. Why the photothermal heating curves of Apt-bioinspired MVs under the same conditions in Figure 2g (from 30 °C to 42 °C) and Supplementary Figure 11 (from 25 °C to 42 °C) have significant differences?

Responses to the Referees' Comments and the Corresponding Revisions

We thank the referees for their positive endorsement and pertinent comments about our paper. We have carefully considered the referees' comments and tried to modify the manuscript accordingly. Our responses and corresponding revisions are as follows:

Response to Reviewer 1:

Comment: I confirm my previous comments: The authors are realized a very good work in the new version of the manuscript with the additional information and new figures more completed. The answers of the comments are good and many comments were added in this new version.

Response: Thanks very much for the reviewer's comments.

Response to Reviewer 2:

Comment: The authors have revised the manuscript as my pointed-out portions. So I feel like to be enough the manuscript for acceptance in your journal.

Response: Thanks very much for the reviewer's comments.

Response to Reviewer 3:

Comment: As I have mentioned before, More data should be provided to prove that such tiny amount and slow degradation rate of BPQDs in vivo can effective increase the concentration of inorganic phosphate. Although the authors have added an experiment, I still think that the concentration of PO_4^{3-} in normal body fluids is very much higher than that degraded from BPQDs. What amount of the BP was used? Even if the increased PO_4^{3-} is just from BP, why such short time NIR light irraddiation in vivo can cause the degradation of PLGA and BP?

Response: Thanks very much for the reviewer's comment. The amount of the BP in the Apt-

bioinspired MVs for bone regeneration is about 1.2 mg, as indicated in the prior response letter. The BP can degrade and produce phosphate ions in physiological environment without NIR light irradiation. In our study, the NIR light irradiation was used for generating heat based on the good photothermal effect of BP.

It is a widely employed method to accelerate bone regeneration by increasing the phosphate ions concentration in bone defect region with phosphorus-rich materials (LeGeros, R. Z. *Chem. Rev.* **2008**, *108*, 4742; Pina, S. *et al. Adv. Mater.* **2015**, *27*, 1143). Hwang *et al.* demonstrated that $\text{Ca}_{18}\text{Mg}_2(\text{HPO}_4)_2(\text{PO}_4)_{12}$ nanoparticles can promote bone regeneration by continuous supply of phosphate and magnesium ions under physiological conditions (Hwang, N. S. *et al. Biomaterials* **2017**, *112*, 31). Varghese *et al.* showed that hydrogels loaded with crystalline calcium phosphates could promote bone repair by creating phosphate and calcium rich microenvironment (Varghese, S. *et al. Tissue Eng., Part A* **2018**, *24*, 1148). Soleimani *et al.* reported that poly(L-lactide) nanofibers loaded with hydroxyapatite had good bone repair capacity (Soleimani, M. *et al. Biomacromolecules* **2010**, *11*, 3118). Many other studies have also reported that phosphorus-rich materials can stimulate bone regeneration (Lin, Y. F. *et al. ACS Appl. Mater. Interfaces* **2017**, *9*, 30437; Nam, K. T. *et al. Adv. Healthcare Mater.* **2016**, *5*, 128; Ouyang, H. W. *et al. Adv. Funct. Mater.* **2014**, *24*, 4473; Leeuwenburgh, S. C. G. *et al. Biomaterials* **2014**, *35*, 2014; Qian, Z. Y. *et al. Biomaterials* **2012**, *33*, 4801; Kaplan, D. L. *et al. Bone* **2008**, *42*, 1226). In our manuscript, we integrated the advantages of aptamer, phosphate ions from BPQDs and heat generated by BPQDs for bone regeneration by constructing the Apt-bioinspired MVs biomineralization system. **The phosphate ions concentration variation experiments (Figure 6l in the manuscript) showed that mice treated with the Apt-bioinspired MVs displayed higher concentration of phosphate ions in the bone defect region than the untreated mice, demonstrating that the Apt-bioinspired MVs can increase the concentration of phosphate ions in the bone defect region.**

The amount of BPQDs in the Apt-bioinspired MVs for bone regeneration, the introduction about using phosphorus-rich materials to provide phosphate ions for bone regeneration, and related reference cited as ref. 21, 22 and 23 were added into the revised manuscript.

Comment: The authors want to show a strategy for Molecular Recognition-Guided Biomineralization. As indicated in Figure 1, a tail vein injection was proposed. And the authors mentioned “cell-targeting aptamer modified bioinspired MVs (denoted as Apt-bioinspired MVs) are designed for promoting cell biomineralization (Fig. 1). The aptamer can guide the bioinspired MVs to targeted cells.” However, the authors choose intracranial filling in the followed animal experiments for bone defect repair. I think tail vein injection of the Apt-bioinspired MVs is required for the demonstration of the Molecular Recognition-Guided Biomineralization.

Response: Thanks a lot for the reviewer’s comment. Supplementary Figure 22 in the original supporting information showed the tail vein injection of the Apt-bioinspired MVs for molecular recognition-guided biomineralization. The Apt-bioinspired MVs displayed better biomineralization performance than the bioinspired MVs without aptamer functionalization, suggesting that aptamer can guide the bioinspired MVs to targeted cells for enhanced biomineralization performance.

The detailed description about Supplementary Figure 22 was added into the supporting information of the revised manuscript.

Comment: Why the photothermal heating curves of Apt-bioinspired MVs under the same conditions in Figure 2g (from 30 °C to 42 °C) and Supplementary Figure 11 (from 25 °C to 42 °C) have significant differences?

Response: Thanks to the reviewer for his/her comment. The difference between Figure 2g and Supplementary Figure 11 is due to the different ambient temperature in conducting these two experiments.

REVIEWERS' COMMENTS:

Reviewer #3 (Remarks to the Author):

The answers of the comments 2 and 3 are good and some results were added in this new version. However, in the reply for comment 1, I am still confused. In the answer, the authors claim that the BP can degrade and produce phosphate ions in physiological environment without NIR light irradiation. I agree with that bare BPs can degrade. However, in the experiments, the BPs in Apt-bioinspired MVs are protected by the PLGA shells. Just as the authors' discription in the paper "the bare BPQDs degrade rapidly in humid environment. However, the photothermal performance of the Apt-bioinspired MVs decreases much slowly, confirming their good stability in humid environment." Similar results are shown in Supplementary Figures 8-10. So, since the Apt-bioinspired MVs are stable, why they "can degrade and produce phosphate ions in physiological environment." I am still confused that the BPQDs in Apt-bioinspired MVs are stable or not stable? If they are stable, why BPs in Apt-bioinspired MVs can degrade and release so many amounts of phosphate ions? Since it is the basis of the mechanism proposed by the paper, the authors should explain it clearly for the readers. If the authors can explain it clearly in the revised manuscript, I think the paper can be published.

Responses to the Referees' Comments and the Corresponding Revisions

We thank the referees for their positive endorsement and pertinent comments about our paper. We have carefully considered the referees' comments and tried to modify the manuscript accordingly. Our responses and corresponding revisions are as follows:

Response to Reviewer 3:

Comment: The answers of the comments 2 and 3 are good and some results were added in this new version. However, in the reply for comment 1, I am still confused. In the answer, the authors claim that the BP can degrade and produce phosphate ions in physiological environment without NIR light irradiation. I agree with that bare BPs can degrade. However, in the experiments, the BPs in Apt-bioinspired MVs are protected by the PLGA shells. Just as the authors' discription in the paper "the bare BPQDs degrade rapidly in humid environment. However, the photothermal performance of the Apt-bioinspired MVs decreases much slowly, confirming their good stability in humid environment." Similar results are shown in Supplementary Figures 8-10. So, since the Apt-bioinspired MVs are stable, why they "can degrade and produce phosphate ions in physiological environment." I am still confused that the BPQDs in Apt-bioinspired MVs are stable or not stable? If they are stable, why BPs in Apt-bioinspired MVs can degrade and release so many amounts of phosphate ions? Since it is the basis of the mechanism proposed by the paper, the authors should explain it clearly for the readers. If the authors can explain it clearly in the revised manuscript, I think the paper can be published.

Response: Thanks very much for the reviewer's comment. We realized that the expression "the photothermal performance of the Apt-bioinspired MVs decreases much slowly, confirming their good stability in humid environment" was not accurate. According to the reviewer's valuable comments, we changed the expression to "the photothermal performance of the Apt-bioinspired MVs decreases much slowly, suggesting that BPQDs in the Apt-bioinspired MVs have better stability than the bare BPQDs". Also, we quantified the amount of the produced phosphate ions from Apt-bioinspired MVs over time (Figures 1 and 2 below). Figures 1 and

2 below were added into the supporting information of the revised manuscript.

Previous studies show that polymer encapsulation or modification can increase the stability of BPQDs in humid environment, that is, the encapsulated BPQDs or surface-modified BPQDs show slower degradation rate than bare BPQDs. However, the encapsulated BPQDs also degrade over time to produce phosphate ions because the BPQDs can still get contact with water molecules (Shao, J. *et al. Nat. Commun.* **2016**, *7*, 12967; Tao, W. *et al. Adv. Mater.* **2017**, *29*, 1603276; Sun, Z. *et al. Angew. Chem., Int. Ed.* **2015**, *127*, 11688; Wang, H. *et al. Small* **2018**, *14*, 1702830). In our manuscript, we showed that BPQDs in PLGA displayed slower degradation rate than bare BPQDs, suggesting that PLGA encapsulation can increase the stability of BPQDs in humid environment (Figure 2g). *In vitro* and *in vivo* degradation experiments further showed that BPQDs in Apt-bioinspired MVs degraded to produce phosphate ions over time (Supplementary Figure 16 and Figure 6l).

According to the reviewer's kind comment, we have quantified the amount of the produced phosphate ions over time in the *in vitro* (Figure 1 below) and *in vivo* (Figure 2 below) degradation experiments. Results showed that the slowly degraded BPQDs in Apt-bioinspired MVs could generate large amounts of phosphate ions, showing that the Apt-bioinspired MVs can provide phosphate ions for bone regeneration. Figures 1 and 2 below was added into the supporting information of the revised manuscript as Supplementary Figure 16b and Supplementary Figure 23, respectively.

Figure 1. The amount of produced phosphate ions in the *in vitro* degradation tests of Apt-bioinspired MVs.

Figure 2. The amount of produced phosphate ions in the *in vivo* degradation tests of Apt-bioinspired MVs.